# Intracavitary Applications for CEUS in PTCD

**DOI:** 10.3390/diagnostics14131400

**Published:** 2024-06-30

**Authors:** Evelina G. Atanasova, Christo P. Pentchev, Christian P. Nolsøe

**Affiliations:** 1Faculty of Medicine, Medical University of Sofia, 1431 Sofia, Bulgaria; 2Clinic of Gastroenterology, “St. Ivan Rilski” University Hospital, 1431 Sofia, Bulgaria; 3Centre for Surgical Ultrasound, Department of Surgery, Zealand University Hospital, 4600 Køge, Denmark; cpn@regionsjaelland.dk; 4Institute for Clinical Medicine, University of Copenhagen, 2200 Copenhagen, Denmark

**Keywords:** percutaneous transhepatic cholangiodrainage, intracavitary, contrast enhanced ultrasound, biliary obstruction

## Abstract

Intracavitary contrast-enhanced ultrasound is widely accepted as a highly informative, safe, and easily reproducible technique for the diagnosis, treatment, and follow-up of different pathologies of the biliary tree. This review article describes the diverse applications for CEUS in intracavitary biliary scenarios, supported by a literature review of the utilization of the method in indications like biliary obstruction by various etiologies, including postoperative strictures, evaluation of the biliary tree of liver donors, and evaluation of the localization of a drainage catheter. We also provide pictorial examples of the authors’ personal experience with the use of intracavitary CEUS in cases of PTCD as a palliative intervention. Intracavitary CEUS brings all the positive features of US together with the virtues of contrast-enhanced imaging, providing comparable accuracy to the standard techniques for diagnosing biliary tree diseases.

## 1. Introduction

Ultrasonography (US) is a proven diagnostic method with a long list of generally accepted advantages. It is a non-ionizing and cheap method that can be performed at the patient’s bedside, is easily reproducible, and, in addition, is a modality from which physicians gain ample information for various organs [1]. The method is commonly utilized as a diagnostic tool for biliary tree pathology, because of its satisfactory evaluation [2]. In cases of focal lesions, contrast-enhanced ultrasound (CEUS) may provide additional information, often leading to conclusive diagnosis [3]. A CEUS examination is relatively inexpensive, easy to perform, and non-invasive in nature except for the intravenous access, and it often provides supplementary insights of decisive character [2]. Contrast agents (UCAs), used for CEUS, contain microbubbles of gas enveloped by protein, lipid, or polymer [4]. The most commonly used UCA in Europe, SonoVue ^®^, consists of sulphur hexafluoride, enveloped in a shell of phospholipids. The excretion of the gas is pulmonary, while the excretion of the phospholipid component is by the liver [5]. Their lack of renal impact makes them appropriate for use in kidney impairment [6]. CEUS can detect blood flow in far smaller vessels, compared to Doppler US, i.e., diameters of 40 μm and 100 μm, respectively [2].

At low acoustic pressure, microbubbles resonate, forming non-linear signals that can be detected on the US, and after removing the linear signals from the surrounding tissue, a contrast image is formed, which can be followed in dynamics. These features render CEUS an attractive alternative to fluoroscopic imaging for endocavitary use due to its non-ionizing nature and, in addition, its non-iodinated contrast agent formulation [7]. Much smaller amount of contrast is needed in endocavitary CEUS because the substance is not washed out with the blood stream and thus remains in the cavity longer compared to when applied in the systemic circulation [8,9]. UCA is considered safe, with rarely reported adverse events, when used intravenously, which may indicate even a safer profile in intracavitary applications [10,11]. The introduction of UCAs into physiological or pathological cavities of the body is very useful for evaluation of its inner structure and shape, potential fistula, position of a drainage system, and patency of a hollow organ or duct (e.g., fallopian tubes, biliary system, or reflux detection), etc. [12,13,14,15,16,17,18,19].

This is also embedded in the EFSUMB guidelines and recommendations for the clinical practice of CEUS in non-hepatic applications from 2018, which assume that injection of ultrasound contrast media into physiological or non-physiological cavities is aiding in managing different clinical problems: identification of needle or catheter position, delineation of any cavity or duct, or support tracking of a fistula. No fixed dose of US contrast agent is suggested for intracavitary use, and the range varies between 0.1 and 1 mL of SonoVue™ (or a few drops) diluted in ≥10 mL of 0.9% normal saline [20].

These indications are very important since they provide a potential alternative to contrast fluoroscopy [21]. Microbubbles can be used as echo-enhancers in any non-vascular cavity, which allows detailing and clarification, thus facilitating essential ultrasound-mediated therapy [22]. Among the list of applications of intracavitary CEUS, some of the most important are percutaneous transhepatic cholangiography (PTC), endoscopic ultrasound (EUS) with endoscopic retrograde cholangiography (ERCP), tracing peritoneal-pleural communication, and evacuation of abscesses [20]. Overabundance of UCA leads to lower imaging quality by posterior acoustic shadowing, and it is mandatory to emphasize that a lower dosage of ultrasound contrast agent (UCA) is needed compared with intravenous use [23].

In the case of abscess formation, intravenous UCA delineates the avascular content and helps determine the appropriate access for drainage [23,24].

In the scenario of cholangiography, CEUS can be utilized to identify post-procedural complications such as, e.g., hepato-biliary fistula formation, where contrast enhancement is observed not only in the biliary tree but also in the liver [5]. The method is also applicable in determining the position of a drain and evaluating a biliary obstruction or leakage [13,25,26,27].

In this review article, we aim to offer a nuanced perspective on the highly informative and non-radiative technique for facilitating percutaneous transhepatic cholangiography and drainage (PTCD), whether used independently or in conjunction with fluoroscopy. We seek to highlight the significant advantages of a method capable with great accuracy of identifying biliary stenoses, obstructions, and leaks, ensuring accurate drainage placement at the bedside, without the use of radiation, and allowing for repeated applications. To this end, we provide a comprehensive summary of the available literature on the topic. Additionally, we share our personal experiences with PTCD placement and drainage as a palliative intervention, illustrating the practical utility and efficacy of this method as well as some technical aspects of our practice. 

## 2. Indications for Biliary Decompression

Percutaneous transhepatic cholangiography and drainage (PTCD) is a widely utilized procedure for diagnosing and treating benign and malignant biliary pathology [28,29,30]. As a rule, ERCP is the modality of choice for biliary diagnostic and therapeutic interventions [31]. Endosonographically (EUS)-guided cholangiography and drainage is gaining wide popularity among endoscopists and is practiced at many specialized centers around the world. Indications for PTC and PTCD are cases where an easier and less invasive endoscopic procedure is not possible for diagnosis or treatment [32,33] (e.g., biliary-enteric anastomosis, a Billroth II operation, gastrectomy, hepaticojejunostomy, a Roux-en-Y choledochojejunostomy with a failed afferent limb, duodenal peripapillary diverticula, etc.), or where endoscopic passing through a stenosis has failed [31].

Real-time imaging with ultrasound (US) is useful for the guidance of PTCD (US-PTCD), especially in difficult approaches for non-dilated ducts and left-sided bile duct branches [28,34].

In terms of diagnostics for biliary tree, the available approaches are MRCP, endoscopic ERCP, PTC, and endosonographically guided cholangiography and drainage (EUS-CD). The non-MRI-dependent above-mentioned techniques are used more commonly for therapeutic than diagnostic purposes, because of the availability of magnetic resonance cholangiopancreatography (MRCP). PTC is a second-line procedure in cases of unsuccessfully performed ERCP [35], mainly when an endoscopic approach is not possible. A combined technique between PTC and ERCP is used—the rendezvous technique. PTC is by far preferred by some authors in cases of interventions in the hilar region and in the strictures of the intrahepatic biliary ducts [36,37,38].

The procedure can be utilized in all biliary diseases—benign, malignant (including brachytherapy), and anomalies in the development of the tree [39,40]. Following iatrogenic injury to the biliary tree, the application of PTC is comparable to that of ERCP to determine the anatomy, the location of leakage, and to prove a possible stricture [41]. PTC is helpful for localization and for the placement of percutaneous transhepatic biliary drainage (PTBD). Redirection of bile, facilitation of the localization of the injury, and balloon dilatation of strictures are all possible because of PTBD [42]. The absolute and relative contraindications for PTCD are not different from every other invasive procedure [35].

PTBD in non-dilatated biliary ducts is indicated in symptomatic fistulas (incidence after surgery of the liver, biliary tree, and pancreas: 3–10%) [43,44,45,46,47,48], in cases of bad performance status, long-lasting fistulas, or inaccessible leaks by ERCP. PTBT has a relatively low risk and a considerably positive result. The procedure has proven to be less intricate in dilated ducts than in non-dilated ducts [49,50,51,52,53]. The reason for the absence of dilatation is the presence of leaks, despite the distal stenosis. This leads to the need for smaller catheters to be placed in small-caliber ducts [49,50,53]. Interventions close to the hepatic hilum increase the risk for complications such as haemobilia [54]. It is presumed that draining bile outside rather than into the bowel favors fistula healing and reduces the risk of superinfection [55].

Indications for PTC and PTCD were in all cases where an easier and less invasive endoscopic procedure was not possible for diagnosis or treatment [32,33], or where endoscopic passing through a stenosis had failed. As a rule, the indication for PTC is mainly therapeutic for the decompression of an obstructed biliary tract [31]. Injecting intra-biliary contrast can aid in identifying leaks, strictures, and the position of drains [7].

## 3. Technical Aspects

### 3.1. PTC Technique

The literature holds a variety of different procedures for the placement of PTC drainage catheters, and there may be advantages and disadvantages to one over the other. Our standard procedure utilizes a loop catheter of 40 cm and 6 F to 8 F diameter. The catheter is placed under ultrasound guidance using the Seldinger technique and an initial puncture with a 1.2 mm lumbar needle. When the lumbar needle is visualized with its tip inside selected dilated bile ducts and bile can be aspirated with a syringe, then a 0.035-inch guidewire is inserted, and at this point, if an X-ray is available, it may be advantageous to visualize the passage of the guidewire to deeper bile duct sections. When the guidewire is deemed correctly in position, the lumbar needle is exchanged with the pigtail catheter over the guidewire. Potentially 6 F to 10 F dilators can be used before inserting the catheter; however, when using the Seldinger technique, there is often no need for a dilatator if only the inner needle is taken out of the pig-tail catheter, leaving the outer needle shaft to stabilize the catheter during insertion over the guidewire. When the catheter is supposedly in the correct position, final confirmation is obtained by CEUS cholangiography, which may also be supplemented with X-ray contrast cholangiography. Finally, the catheter is securely fixed to the skin with a suture or dedicated fixture device to ensure inadvertent displacement, and before sending off the patient, the catheter is connected to a drainage bag.

Drainage sets are available in different lengths with a wide range of diameter from 6 F up to even 18 F (the latter being rarely indicated), all featuring side holes (with a distance from the skin of around 7.5 cm). These sets come complete with a connecting tube, sealing cap, and skin plate. The PTCD catheter includes a pigtail end, multiple side holes, and an internal string fixture that holds the pig-tail shape in position. When pulling out the catheter, it is crucial to cut the string to avoid scratching the liver surface, thereby causing potential peritoneal bleeding. Numerous additional treatment options exist, spanning from the rendezvous technique for ongoing drainage via minimally invasive endoscopic methods to the percutaneous placement of a metal stent [31].

Despite the diversity of the published data on the PTCD complication rate, serious adverse events have been observed less in recent years [31]. In a recent observational study by Turan et al., during a 5-year period, a total of 331 patients underwent PTCD, of whom 205 (61.9%) developed PTCD-related complications. Of the patients without a pre-existent infection, 40.6% developed infectious complications, i.e., cholangitis in 26.3%, sepsis in 24.6%, abscess formation in 2.7%, and cholecystitis in 1.3%. Non-infectious complications developed in 34.4%. 30-day mortality was 17.2% [56]. The use of smaller-sized needles is associated with a significantly lower complication rate [57,58]. Important to emphasize is the fact that most of these patients have end-stage malignant disease of a variety of etiologies, and the procedure is most often performed with a palliative purpose.

### 3.2. Intracavitary CEUS Technique

The necessary equipment for intracavitary CEUS into a drainage tube for PTC includes a second-generation ultrasound contrast agent (SonoVue^®^ Bracco, Milan, Italy), a 10 cc syringe, and standard saline solution. The injection is performed directly into the connecting tube. It is important not to use any additional tools with membranes and not from the side arm of a 3-way screwcock (Figure 1).

No accurate and scientifically validated numbers are available, but some authors recommend a very high dilution of US contrast agent (0.1–0.2 mL SonoVue per 20 mL physiologic saline solution) injected into the bile ducts after the initial puncture is performed to confirm proper intraductal placement or identify a possible stenosis [9,20,29,31]. Our personal experience, though, is that a few droplets in 10 or 20 cc will suffice, and care should always be taken to avoid an overabundance of contrast, which will result in the shadowing of deeper structures. It is also very important to use the recommended (as always with CEUS) low mechanical index (less than 0.2). Roberts et al. were among the first to demonstrate the injection of perflutren lipid microspheres (Definity®, Lantheus Medical Imaging, Inc., USA), an ultrasound contrast agent, in various dilutions through an intraoperatively cannulated porcine common bile duct. The utilization of micro-bubble contrast in the biliary system resulted in outstanding visualization of the bile duct, extending down to ducts as small as fifth-order branches and revealing bile ducts measuring less than 2 mm in diameter [59]. In one of the first reported human studies after insertion of the drain, SonoVue^®^ 1 mL was injected, followed by another 5 mL saline flush [60]. According to the experience of Ignee et al. as an approach for diagnostic cholangiography after sufficient local anesthesia and sedation, a suitable biliary duct (mainly in segment 5 or 6) is targeted under US guidance and punctured by means of a 20 G Chiba needle. One to two milliliters of ultrasound contrast agent mixture (again 0.1 mL in 20 mL of physiological saline solution) is injected for correct positioning of the needle in the bile ducts, followed by the insertion of a thin catheter (5 F). In the case of therapeutic PTC placement, there is no need to inject UCA until the end to confirm the correct placement of the catheter. In critically ill patients, a drain (mostly 8–10 F) might be used for external bile drainage. If the procedure results in improvement of the patient’s general condition, the internalization of the drain into the intestine can be performed either via a rendezvous procedure with ERC, PTCD with the application of an internal–external drain, or via the same PTC catheter with the placement of a permanent metal stent. The PTC may simply be left in position in terminal patients if it is placed for palliation to alleviate symptoms from bile duct obstruction such as skin itching, general malaise, or cholangitis. In oncology treatment, the strategy could also be to place PTC in order to bring down elevated bilirubin blood levels, allowing for chemotherapy. In such scenarios, the catheter can be discontinued after treatment if successful opening of the stricture results from the chemotherapy. Another very important utility of the injection of US contrast agent through the drainage tube is the possibility of detecting complications of PTCD like bile leakage, drain dislocation, and fistulas in the pleural cavity [61].

Xu et al. commenced by injecting 1 mL of UCAs into the drainage catheter to initially pinpoint the tip location. They undertook this approach to prevent the overflow signal of UCAs from obscuring the drainage catheter and inundating the entire biliary tract. Once the positions of the drainage catheter were verified, additional UCAs were introduced gradually through the drainage catheter to illustrate the level and extent of obstruction within the biliary tract. In certain patients with severe incomplete obstruction, enhancing the dosage and pressure of the UCA injection could potentially enhance diagnostic accuracy. As a result, the authors proposed a novel PTBD strategy. Rather than employing US-guided PTBD in conjunction with fluoroscopic cholangiography, Xu et al. conducted IB-CEUS concurrently with conventional US-guided PTBD, omitting the need for fluoroscopic cholangiography. This innovative approach offers time- and cost-savings, as the entire procedure can be conducted at the bedside without the need for radiation exposure. It represents a comprehensive “one-station” strategy for PTBD [27].

In a very recent pictorial assay, Chen et al. report their experience with percutaneous ultrasound cholangiography with microbubbles in children with biliary diseases. To disperse sulfur hexafluoride and phosphatide, the authors inject 5 mL of 0.9% sodium chloride solution into the bottle. Subsequently, the microbubble dispersion is mixed with 30 mL of 0.9% sodium chloride solution in a 35 mL injection syringe. The mixed US microbubble contrast agents are then injected into the drainage tube. The distribution of the microbubbles in the biliary system is observed under the contrast imaging mode [62].

The impact of UCAs on the biliary system remains largely unexplored due to the limited literature on the subject. In one of the largest studies on the topic, no instances of chemical cholangitis were reported [27]. However, there is a potential risk of cholangitis associated with overdistention, particularly in dilated systems. Therefore, it is advisable to minimize the amount of contrast injected. Additionally, the long-term effects of sulfur hexafluoride on the biliary epithelium have yet to be thoroughly investigated [63].

In the study of Luao et al., after depicting the level of biliary obstruction, the contrast agent liquid remaining in the bile duct was aspirated to the greatest extent possible. This procedure was undertaken to minimize the risk of any unexpected damage to the bile duct [13].

## 4. Intracavitary Use of CEUS in PTCD in Different Biliary Obstruction Setting (Table 1)

The conventional approach for the placement of an external biliary drainage catheter integrates fluoroscopy with initial sonographic visualization of the dilated bile duct region. In cases of a malfunctioning drainage catheter or if cholestasis parameters remain elevated, it is imperative to evaluate whether the catheter is defective or dislodged, or if there is an internal and/or external obstruction impeding drainage [64]. This may be successfully achieved by injecting a small amount of ultrasound contrast agent through the drainage tube and visualizing it in real time.

**Table 1 diagnostics-14-01400-t001:** Summary of the studies on intracavitary/intrabiliary CEUS during or post-PTCD.

Study	N	M/F	Indication	Groups	Success rate	Complications	Approach Side	Repeated Injections	Fluoroscopy Comparison	Bilateral Approach	Determining Level of Obstruction	Accuracy
Ignee et al. 2009	8	NA	Klatskin *n* = 5 (50%); Distal bile duct Ca *n* = 1; Ca pancreas *n* = 1; Chronic obstructive Pancreatitis with pseudocyst *n* = 1; Trauma of common bile duct *n* = 1	NA	*n* = 8 (100%)	displacement of the drain *n* = 1	Rightsided *n* = 6 (75%); Median subcostal *n* = 2 (25%)	*n* = 2	Yes, but for guidance after stenosis is identified	No	Not indicated	NA
Zheng et al. 2010	12	10/2	Evaluation of biliary tree in living liver donors	Normal biliary pattern biliary variations	*n* = 12 (100%) for adequate evaluation of biliary tree anatomy	no	Canulation of cystic duct and CBD intraoperatively	No	Yes-intraoperative cholangiography	NA	NA	Excellent for first order branches
Luyao et al. 2011	58	37/21	Hilar cholangiocarcinoma *n* = 19; HCC with invasion of common hepatic duct *n* = 2; Hilar cholangitis *n* = 2; Periampullary tumor *n* = 8; Common bile duct stone *n* = 10; postoperative stricture *n* = 2	Hilar obstruction Extrahepatic obstruction	Not indicated	PUSC—pain in the right upper quadrant *n* = 3 PTC—epigastric pain *n* = 25	Not indicated	No	Yes—after PUSC	*n* = 8	PUSC—hilar 100% accuracy; PUSC—CBD obstr. 93.8% accuracy; PTC—hilar 100% accuracy; PTC—CBD obstr. 100% accuracy	For level of obstruction: PUSC 96.6%; PTC 100% For cause: PUSC 93.1%; PTC 79.3%
Xu et al. 2012	80	61/19	Localize the drainage catheter; Localize the distal tip of catheter; Evaluate level and degree of biliary obstruction	Intrahepatic obstr. *n* = 44; Extrahepatic obstr. *n* = 36; Complete *n* = 56; Incomplete *n* = 24	*n* = 80 (100%)	4 catheters not properly placed, which required reposition	Not indicated	Yes	Yes; FC *n* = 68; CTC *n* = 12	NA	100% accuracy (extrahepatic/intra hepatic)	96.3% (77/80) for complete/incomplete; 100% for tip location
Chopra et al. 2012	12	7/5	Evaluation of the biliary tree via T-tube after liver transplantation	Intrahepatic; Extrahepatic	Comparable to fluoroscopy pathology found in 4 pts anastomotic stenosis *n* = 1; Delayed duodenal outflow *n* = 2; Anastomotic leakage *n* = 1	not indicated	Via postoperative T-tube	No	Yes fluoroscopy superior in identifying anastomotic stricture and leakage	NA	CEUS inferior in visualization of extra hepatic bile ducts	NA
Ignee et al. 2015	38	25/13	Pancreas adenocarcinoma *n* = 11; CBD stone *n* = 6; Klatskin tu *n* = 6; Inflammatory stricture *n* = 5; Pancreatic meta *n* = 2; Lymph node meta *n* = 3; Duodenal Ca *n* = 2 ICC *n* = 1; Neuroendocrine Ca of papilla *n* = 1; IPMN *n* = 1	Hilar (above cystic duct); Extrahepatic; Complete; Incomplete	100%	subcutaneous hematoma *n* = 1 catheter dislodgement *n* = 2 pleural effusion and peural- peritoneal fistula *n* = 1	Rightsided *n* = 33; Median left hepatic *n* = 5	Yes (several) plus the day after the intervention	Yes (after CEUS contrast)	NA	Hilar obst. *n* = 8; Extrahepatic obstr. *n* = 30	97.4% for degree Incomplete 9/10

### 4.1. Intracavitary CEUS in PTCD Due to Benign or Malignant Biliary Obstruction

In this particular utilization, the purpose of intracavitary CEUS is to identify the site of obstruction or possible stenosis, as well as to locate the catheter tip and confirm the successful placement of the bilary drainage catheter. There are several studies that report excellent results in this direction.

The initial experience is by Ignee et al. on a small group of patients with various obstructive biliary pathologies and intrahepatic duct dilatation of 2 mm and more. A right-sided approach was used in the majority of the cases. The initial experience with the CEUS-PTCD intervention was successful in all patients for the identification of stenosis. Dislodgement of the catheter occurred in 1 patient, and the catheter was re-inserted. In this study, fluoroscopy was used, but only for guidance [60].

In a larger cohort of 58 patients with various obstructive biliary pathologies, percutaneous ultrasound cholangiography (PUSC) was compared with percutaneous transhepatic cholangiography (PTC). For 8 patients, the bilateral approach was necessary for drainage catheter placement. The patients were divided according to the level of obstruction into hilar obstruction and extrahepatic obstruction groups, respectively. The strength of PUSC was in defining the level of obstruction as hilar with an accuracy of 100%. The determination of CBD obstruction was weaker, with an accuracy of 93.8%. The accuracy of PUSC in determining the level of obstruction was 96.6% (56/58). The accuracy of PTC in both groups was 100% (*p* = 1.000, *p* = 0.492, respectively). In general, the accuracy of PUSC in determining the cause of obstruction was 93.1% (54/58) while that of PTC was 79.3% (46/58). The difference between them was statistically significant (*p* = 0.031) [13].

Xu et al. conducted a study performing intracavitary CEUS during PTCD in the largest number of patients (*n* = 80) so far. The indications for performing the intrabiliary CEUS were to localize the drainage catheter, to localize the distal tip of the catheter, and/or to evaluate the level and degree of obstruction. The US contrast agent injections were performed in a two-step manner: the first injection to track the distal tip of the catheter and the second injection to evaluate the obstruction of the biliary tract. Four catheters were not properly placed and were repositioned. The intervention was successful in all 80 patients (100%). CT, or fluoroscopy guided cholangiography, was used as a reference method. The patients were defined as having intrahepatic or extrahepatic obstruction according to the level of obstruction and as complete or incomplete, according to the degree of obstruction. The authors report excellent accuracy for intrabiliary CEUS for the determination of the level of obstruction and 96.3% accuracy for detecting the degree of obstruction. The accuracy for locating the proper position of the tip of the catheter was 100% [27].

As a continuation of their initial experience, Ignee et al. conducted another study for extravascular use of CEUS in a larger patient cohort (*n* = 38) with various primary or secondary biliary obstructive pathologies during PTCD after failed or potentially impossible endoscopic retrograde cholangiography (ERC). The intervention was performed in several stages. The diluted US contrast agent was injected before the X-ray contrast agent. In the first stage, both contrast agents were injected to confirm the intraductal position of the needle and to detect the level of obstruction. Then repeated injections were necessary to complete the PTCD, and in cases of malignant stenosis, they were indicated for catheter or stent position confirmation. A right-sided approach was used in the majority of the cases. The puncture and catheter insertion were successful in all patients (100%). Fluoroscopy was used as the standard method. The accuracy of extravascular CEUS was 100% for correctly diagnosing the level of obstruction as hilar or extrahepatic and 97.4% for defining the degree of obstruction (complete/incomplete) [17].

In an effort to elucidate the causes of biliary obstruction, Müller et al. reported their experience with a combination of endocavitary and intravenous administration. This combined technique appears to enhance the quality of diagnostic ultrasound imaging, allowing for precise visualization of bile duct tumors, which have traditionally been challenging to detect [65].

### 4.2. Intracavitary CEUS in PTCD after Liver Transplantation or Abdominal Surgery

Biliary tract complications following liver transplantation are a frequent and often complex issue. In this context, native ultrasound is limited in its ability to display a non-dilated biliary tree. However, with the injection of intraductal contrast media, contrast-enhanced ultrasound (CEUS) facilitates selective imaging of the biliary tree. This has been demonstrated through the intraoperative injection of microbubbles via the cystic duct and their application through PTCD [66].

A study by Zheng et al. included 12 living liver donors for evaluation of the biliary tree. The approach was via cannulation of the cystic duct and common bile ducts intraoperatively. The intervention was successful for adequate evaluation of the biliary anatomy in all patients, and as a result, in three of the donors were diagnosed with biliary variations, and the rest of the donors had a normal biliary pattern. No complications were described. Intraoperative cholangiography was used for comparison. The accuracy of 3D ultrasound cholangiography was excellent for first- and second-order biliary branches [67].

Chopra et al. evaluated 12 patients after orthotopic liver transplantation with side-to-side biliary anastomosis with intracavitary CEUS for assessment of postoperative complications. On the fifth postoperative day, they performed conventional cholangiography and CEUS cholangiography via a T-tube placed intraoperatively. Both techniques showed a similar success rate. Pathology was found in 4 patients—one anastomotic stenosis, one anastomotic leakage, and delayed duodenal outflow in 2 cases. Fluoroscopy showed superiority in detecting anastomotic stricture and leakage. Along with the detection of pathology, the visualization of the extra- and intrahepatic bile ducts was assessed. The results showed that CEUS was inferior in the visualization of extrahepatic bile ducts [66].

Two interesting cases of intracavitary CEUS applications are supporting this valuable application (Table 2).

The first case is of a male patient with bile leakage after T-tube removal. The T-tube was placed during cholecystectomy and CBD exploration. After the operation, the patient developed mild fever, jaundice, and abdominal pain, which indicated the performance of US-guided PTCD via a right-sided approach. After an injection of a diluted US contrast agent, a leakage of the contrast from the common bile duct was observed and confirmed by a percutaneous transhepatic cholangiogram [25]. This report clearly demonstrates the utility of intracavitary CEUS for managing postoperative biliary complications in routine abdominal surgery.

The second case is of a young male patient with postoperative biliary complications after liver transplantation for familial amyloid polyneuropathy type 1. On the 35th post-transplant day, a PTCD was performed due to a continuing bile leak at the hepaticojejunostomy site. Seven days after the biliary drain placement, intermittent haemobilia occurred. A CEUS cholangiogram was performed, and after demonstration of the correctly sited distal aspect of the drain within the hepaticojejunostomy with leakage around the liver surface at the site of proximal percutaneous insertion, the presence of an occult biliary-arterial fistula was identified. The latter was confirmed by a conventional tubogram [29].

### 4.3. Various

Although still “off-label”, a well-known application of the CEUS in children, is the administration of microbubble-based US contrast agents for real-time evaluation of their distribution in body cavities [68]. Chen et al. demonstrated the successful utilization of intracavitary CEUS in children with biliary diseases. The authors present their experience in performing percutaneous ultrasound cholangiography and the use of sulfur hexafluoride microbubbles as contrast agents in children with neonatal hepatitis, biliary atresia, choledochal cysts with pancreaticobiliary malfunction, and postoperative complications after hepatobiliary surgery [62].

## 5. Pictorial Examples

We present three cases of personal experience where intracavitary CEUS during or after PTCD played a pivotal role and provided a decisive solution to a critical clinical scenario.

The first case was a palliative PTCD in a patient with terminal-stage gallbladder cancer that spread into the intrahepatic bile ducts (Figure 2). An 8 Fr pigtail catheter was inserted in the left liver lobe for biliary decompression. Three months later, the patient presented with a non-functioning PTC drain. Due to the fact that the injection of a diluted US contrast agent (SonoVue) was initially impossible, a higher pressure on the syringe’s plug was exerted. A possible clot was the reason for the obstruction of the drainage tube, which was overcome. Then the diluted US contrast agent entered the biliary ducts unhindered and confirmed the proper position of the patent drainage tube.

The second case was a patient with carcinoma of the antrum of the stomach with malignant infiltration of the pancreatic head and compression of the biliary tract (Figure 3). The biliary drainage was placed for palliative purposes. The pictures show an intracavitary injection of a diluted ultrasound contrast agent into a biliary drainage catheter inserted in the left liver lobe.

The third case was a female patient with an inoperable Klatskin tumor (Figure 4). ERCP showed >90% malignant stenosis of ductus choledochus at the hilum with a length of 20 mm. The patient was referred for PTCD, and the proper drainage catheter position in the bile ducts of the left liver lobe was confirmed by intracavitary CEUS:

## 6. Discussion

Since the approval of the use of US contrast agents more than 20 years ago in some parts of the world [69], CEUS has proven its excellent diagnostic accuracy comparable with contrast-enhanced CT or MRI, especially in focal liver lesions [70,71,72,73,74,75,76,77]. In addition to this well-accepted usage, endocavitary CEUS represents a dynamic modality assessment in real time and can provide high-quality additional information to the US without further considerations, such as the usage of iodinated contrast and radiation [7].

The current review and our personal experience, including the pictorial cases presented herein and all of the available studies for intracavitary application of CEUS in PTCD, present excellent examples of the benefits and added information from this novel technique. All studies cited in this review show excellent accuracy of intracavitary CEUS for determining the level of obstruction, locating the drainage catheter and its distal tip, identifying biliary leaks, and, in addition, very good ability to define the degree of obstruction as well as delineate the biliary tree anatomy in living liver donors. All these clinical scenarios represent excellent indications and demonstrate the non-inferiority of intracavitary CEUS compared to conventional fluoroscopy.

This relatively new application of US contrast agents into a cavity or drainage catheter is gaining more and more popularity. It is cost-effective, available for bedside use, and possesses all of the advantages of intravascular use, i.e., an excellent safety profile, no nephrotoxicity, and the avoidance of ionizing radiation. Additionally, a very important advantage is the opportunity for repeated examinations during the intervention and for follow-up. This method could prove valuable in managing post-transplantation biliary complications, as well as biliary complications arising in routine abdominal surgery. 

Despite these advantages, there are some limitations to the intracavitary CEUS. First of all, like all clinical ultrasound, it is operator dependent and a lower quality of visualization must be expected in obese patients and in the presence of abundant bowel gas [7]. Additionally, more subtle alterations of the bile ducts, such as those seen in primary sclerosing cholangitis, are not depicted sufficiently [65]. Another still-unexplored area is the investigation of the chemical effect of the US contrast agent on the biliary epithelium. So far, however, there are no reported cases of chemical cholangitis or complications due to increased intraductal pressure after intrabiliary administration of US contrast agents. Another limiting factor is that US contrast agents are not widely available in some parts of the world.

As the number of studies on the topic is very limited and some report small numbers of patients with a high variation in obstructive biliary pathology, further randomized studies with larger patient cohorts and more homogeneous biliary pathology are warranted.

In the future, there may be a focus on developing materials for extravascular contrast-enhanced ultrasound imaging, with an emphasis on using specific compounds such as cell-penetrating peptides. For instance, a disulfide-bridged cyclic RGD (Arg-Gly-Asp) peptide, named iRGD (internalizing RGD), which is a tumor-homing peptide with high affinity and specificity for a certain integrin, could be utilized to construct targeted materials. Integrating iRGD peptide into materials and thereby potentially enhancing penetration of blood vessels and the extracellular matrix, facilitating accumulation, and increasing the likelihood of enhanced imaging [78,79]. Proteins incorporating the Arg-Gly-Asp (RGD) attachment site, along with integrins functioning as their receptors, represent a fundamental recognition system crucial for cell adhesion [80].

Additionally, there are strategies to further increase the accumulation of targeted materials and enhance the probability of improved imaging through the use of NO-releasing agents [81].

There has been a surge of interest in utilizing microbubbles as carriers for drugs, aiming to deliver them to specific sites and achieve localized release through the disruption of microbubbles using high-frequency ultrasound waves. This localized release strategy enhances drug efficiency while concurrently reducing systemic side effects. Particularly notable is its success in facilitating the targeted release of chemotherapeutic agents, effectively mitigating their systemic adverse effects [82].

Another future direction of research is oral contrast-enhanced ultrasound for delineating a fistula, as there are reports of the stability of UCA within the stomach despite acidic conditions [83].

## 7. Conclusions

The intracavitary CEUS has proven to be comparable to fluoroscopy in aiding the PTCD procedure, delineating the normal biliary anatomy, or detecting pathology. It is safe to use, radiation free, reproducible, and could be performed several times during the intervention and for follow-up. Whenever available, intracavitary CEUS should be on the list of physicians’ arsenals to address various diagnostic and therapeutic challenges assisted by imaging modalities.

## Figures and Tables

**Figure 1 diagnostics-14-01400-f001:**
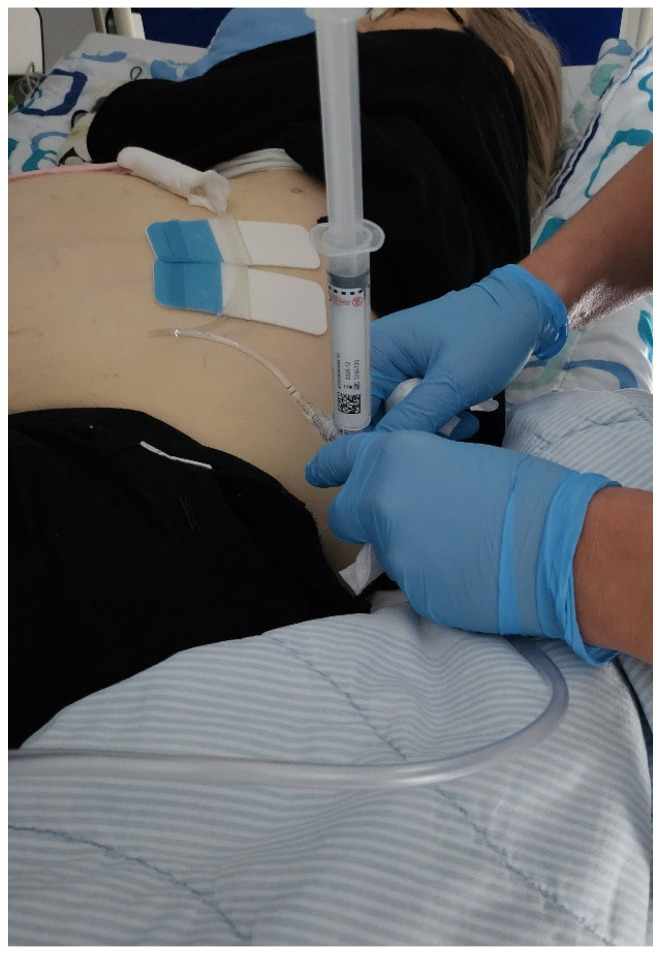
The US contrast must be applied directly into the connecting tube, not through a “chimney” or devices with a membrane and not from an angle, i.e., not from a side-arm of a 3-way screwcock.

**Figure 2 diagnostics-14-01400-f002:**
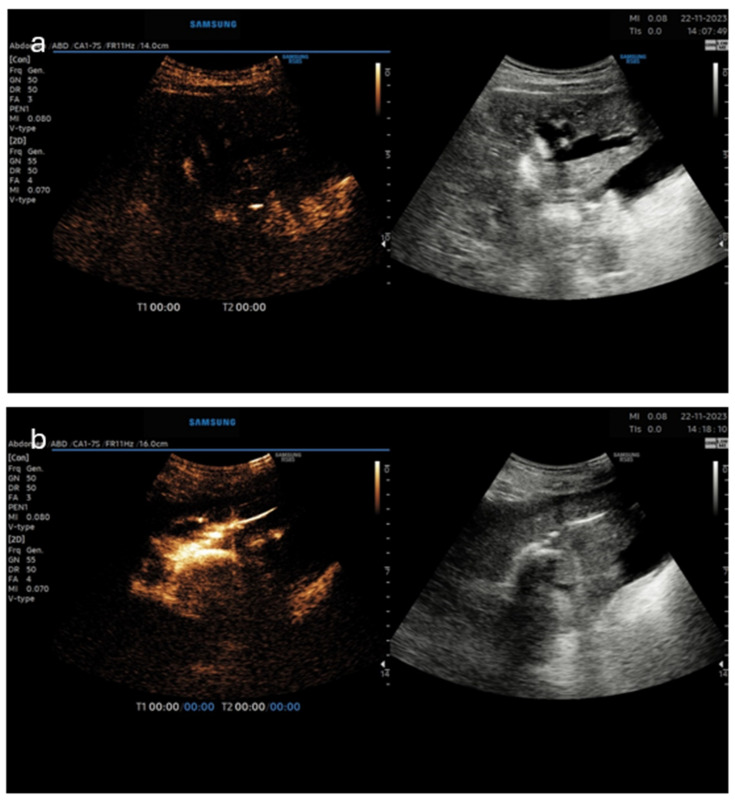
(Dual image—**left** CEUS image, **right**—B-mode). Panel (**a**)—prior to US contrast injection demonstrating markedly dilated intrahepatic bile ducts in the left liver lobe; panel (**b**)—biliary decompression and filling of the biliary catheter and bile ducts after US contrast injection.

**Figure 3 diagnostics-14-01400-f003:**
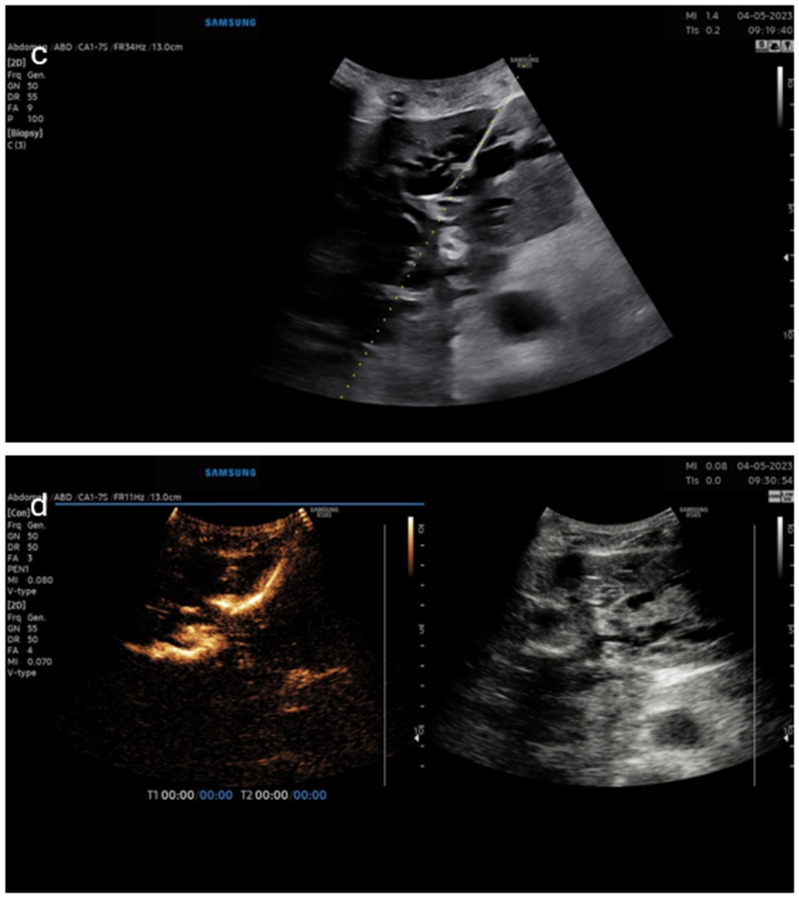
Panel (**c**)—B-mode image of left-sided puncture of the dilated bile ducts; panel (**d**)—(dual image—**left** CEUS image, **right**—B-mode) for confirmation of proper catheter position.

**Figure 4 diagnostics-14-01400-f004:**
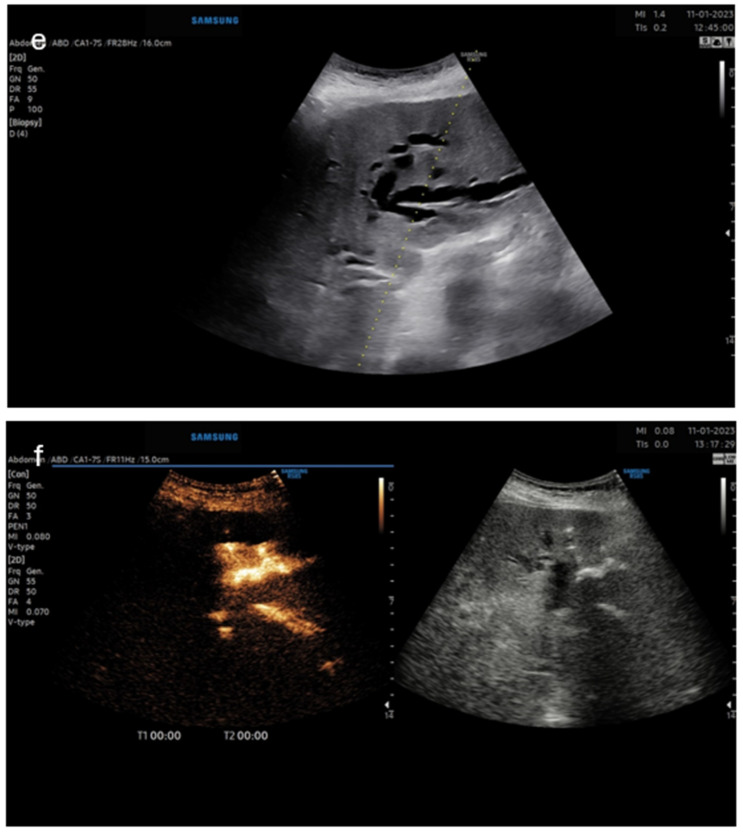
Panel (**e**)—B-mode image of planning the puncture site; panel (**f**)—dual image—**left** CEUS image, **right**—B-mode.

**Table 2 diagnostics-14-01400-t002:** Case reports on intracavitary/intrabiliary CEUS during or post-PTCD.

Authors	Mao	Daneshi
**N**	1	1
**M/F**	M	M
**Indication**	Bile leakage after T-tube removal after cholecystectomy and CBD exploration	Liver transplantation for familial amyloid polyneuropathy type 1—postoperative biliary-arterial fistula
**Successful**	yes	yes
**Complications**	None	Hemobilia after internal–external drainage catheter placement
**Approach side**	Right sided segment VI	Right sided
**Repeated injections**	No	No
**Fluoroscopy**	Yes (for confirmation)	Yes (prior drainage and after CEUS cholangiogram)
**Bilateral approach**	No	No
**Determining level**	Yes (bile leakage from CBD)	Yes (microbubbles at right hepatic artery branch)

## Data Availability

No new data were created or analyzed in this study. Data sharing is not applicable to this article.

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
