# Peer review of "Intracavitary Applications for CEUS in PTCD"

_diagnostics, 2024, doi:10.3390/diagnostics14131400_

Round 1
Reviewer 1 Report
Comments and Suggestions for Authors
1. Abstract line 20 - . One of the studies found a statistically significant difference in the diagnostic accuracy of PTC and PUSC. - Well, you have to first make up the mind if you want this a narrative review, or a retrospective study of your cases/patients or a literature review. It is unacceptable to quote other papers results in your results!
2. Abstract line 22 - "proving comparable accuracy to the listed techniques for diagnostics of biliary tree diseases." - Well, did you calculate the sensitivity, specificity and accuracy? to claim that this is comparable? And what do you claim as 'listed techniques"? For example, do you consider the endoscopic spyglass as a listed technique?
3. Introduction is more like an essay on CEUS and with 52 citations and 2 pages, it is too much heavy. It should be trimmed by about 50% with what is known, what is the gap, how does your study plug the evidence gap and what are the study aims. I do not see any aim or purpose stated in the introduction section.
4. The Table formatting is inappropriate and it cannot be spanning 2 pages. Some editing is necessary to adjust it in 1 page.
5. The result section actually described each study that you have tabulated. Again, it is confusing to readers as you have conducted a kind of systematic review (but without a PRISMA) and without any structure of literature search, you have reported a narrative review, and you combine your patient data. You add complications to method section by also tabulating 2 case reports. I find this haphazard reporting. I suggest you edit this as a retrospective study and enrich the discussion segment with your literature review. In current format, the paper lacks aims, is vague, is too detailed and without a purpose and structure. Alternatively, you make it a pictorial essay and omit the literature review section in such details.
Thanks
Comments on the Quality of English Languagenil
Reviewer 2 Report
Comments and Suggestions for Authors
The manuscript outlines a literature review on the intracavitary application of ultrasound contrast material (UCA) in evaluating hepatobiliopancreatic complications and inserting percutaneous biliary drainage (PTCD).
Furthermore, the authors made a technical presentation of some cases of intracavitary CEUS application.
The article has a good idea. It describes a method that is not very widespread but has good potential. It is economical, bedside, repeatable, and without radiation.
However, it may be helpful to make some clarifications:
- the review does not follow any protocol typically adopted for reviews, even if not systematic (e.g., PRISMA)
- the articles collected in the review are heterogeneous in the indications and methods of application of intracavitary CEUS: the authors could group the cases of the articles according to a criterion (execution method, indication, etc.)
- in the literature there are no statistically valid comparisons between intracavitary CEUS and other currently recognized methods (such as cholangiography); comparing the methods could be helpful.
Overall, the article presents an interesting and underutilized topic, which could find increasingly more space in post-transplant hepatobiliopancreatic management.
Comments on the Quality of English LanguageThe quality of English language is sufficient
Round 2
Reviewer 1 Report
Comments and Suggestions for Authors
good amount of edits are made